# Artificial Intelligence for Prognosis of Gastro-Entero-Pancreatic Neuroendocrine Neoplasms

**DOI:** 10.3390/cancers17121981

**Published:** 2025-06-13

**Authors:** Elettra Merola, Giuseppe Fanciulli, Giovanni Mario Pes, Maria Pina Dore

**Affiliations:** 1Dipartimento di Medicina, Chirurgia e Farmacia, University of Sassari, Viale San Pietro 43, 07100 Sassari, Italy; gfanciu@uniss.it (G.F.); gmpes@uniss.it (G.M.P.); mpdore@uniss.it (M.P.D.); 2Baylor College of Medicine, One Baylor Plaza Blvd, Houston, TX 77030, USA

**Keywords:** GEP-NENs, artificial intelligence, machine learning, deep learning, prognostic models, survival

## Abstract

Gastro-entero-pancreatic neuroendocrine neoplasms (GEP-NENs) are highly heterogeneous, and complicated by risk stratification and therapy selection. Artificial intelligence (AI) can examine large clinical, imaging, and pathological datasets to obtain more accurate survival predictions and support tailored treatment. Early results are promising but derive mainly from small, retrospective cohorts without robust external validation, with ethical issues still persisting. This narrative review investigates current AI-based prognostic models in GEP-NENs, as well as their efficacy in predicting survival and optimizing patient management.

## 1. Introduction

Over the past two decades, comprehensive population-based registries have shown that gastro-entero-pancreatic neuroendocrine neoplasms (GEP-NENs) are far more common than previously believed. In several Western countries, their age-standardized incidence has more than doubled since 2005. This increase has mainly been attributed to improvements in cross-sectional imaging modalities, including computed tomography (CT) and magnetic resonance imaging, as well as to the widespread adoption of screening endoscopy and refinements in histopathological criteria [1].

Advances in diagnostic techniques have significantly deepened our understanding of the biological diversity among GEP-NENs. These neoplasms are now recognized to exist along a broad biological continuum. According to the World Health Organization (WHO) classification, well-differentiated neuroendocrine tumors (NETs), often characterized by an indolent clinical course, occupy one end of this spectrum. At the opposite end are poorly differentiated, high-grade neuroendocrine carcinomas (NECs), which are typically aggressive, rapidly progressive, and associated with poorer clinical outcomes[2]. The clinical course of GEP-NENs is not dictated by differentiation grade alone. Rather, several factors contribute to disease behavior and prognosis, such as the Ki-67 proliferation index, primary tumor site, and disease stage [3,4]. Most GEP-NETs express somatostatin receptors (SSTRs), a molecular hallmark that provides the basis for both functional imaging, such as [^68^Ga]Ga-DOTA-positron emission tomography/CT ([^68^Ga]Ga-DOTA-PET/CT), and for SSTR-directed treatments, including long-acting somatostatin analogs (SSAs) and peptide-receptor radionuclide therapy (PRRT) [5,6]. Nevertheless, curative surgery remains an option only for a limited proportion of patients, and systemic therapies must be sequenced thoughtfully over a chronic disease course. International societies, such as the European Neuroendocrine Tumor Society, therefore emphasize the importance of management in experienced multidisciplinary teams to ensure individualized treatment planning, optimal surveillance strategies, and timely transitions between therapeutic modalities [7,8].

In the diagnostic realm, artificial intelligence (AI) has shown considerable promise across various imaging modalities, including endoscopy, CT, and PET/CT. Notably, in endoscopic ultrasonography (EUS), which remains the gold standard for tissue acquisition in pancreatic lesions [9,10], AI models are emerging as complementary tools for differentiating pancreatic masses. These technologies may be particularly useful in cases where biopsy poses a significant risk or where sampling yields are inadequate. However, in the context of pancreatic neuroendocrine tumors (PanNETs), current AI tools remain limited. Specifically, they lack sufficient accuracy to determine Ki-67 proliferation indices or to distinguish NETs G3 from NECs. Consequently, histopathology remains essential for definitive diagnosis in most cases, although AI may play an adjunctive role [11].

Regarding NEN prognosis, AI techniques, particularly machine learning (ML) and deep learning (DL) algorithms, might also bridge relevant gaps. In fact, conventional prognostic models, typically built upon Cox proportional hazards models and anatomical staging systems, explain only a portion of outcome variability in GEP-NEN cohorts. These models may fail to account for complex interdependencies between patient characteristics, tumor biology, imaging phenotypes, and treatments. Emerging evidence suggests that AI can interrogate high-dimensional clinical, imaging, and molecular datasets to capture non-linear interactions that elude classical statistics. Recent ML-based nomograms and random survival forest models have already outperformed standard staging systems in predicting the postoperative survival of GEP-NENs [12,13]. However, most AI studies are limited by retrospective design, single-center data, and relatively small patient cohorts; external validation is uncommon, and issues of transparency, reproducibility, and ethical governance remain unresolved.

This narrative review summarizes the state of AI-based prognostic modeling in GEP-NENs, with the aims of: (i) describing the emerging tools for survival prediction and their potential applications to patient management; (ii) discussing the methodological and regulatory challenges that must be addressed before these technologies can be integrated into routine clinical practice.

## 2. Methods

We conducted a comprehensive search of the PubMed database on 26 March 2025 to identify relevant studies for this review. The search strategy employed the following Boolean string: (“artificial intelligence” OR AI OR “deep learning” OR “machine learning”) AND (neuroendocrine AND (tumor OR carcinoma OR neoplasm)). The process of study selection is detailed in the PRISMA flow diagram shown in Appendix A [14]. The initial query returned 1809 results. To ensure the analysis was both current and technically meaningful, we restricted inclusion to original research articles published from 2020 onward that presented primary data on AI-based methods for the prognostication of GEP-NENs. We excluded duplicate records, non-original studies such as reviews or editorials, and articles where AI was not a central component of the research framework. Additionally, we manually screened the reference lists of all included articles to identify further relevant publications that may have been missed in the primary search.

Language polishing and minor stylistic adjustments were assisted by an AI writing tool (ChatGPT-4o); all substantive content and interpretations remain the authors’ own.

## 3. AI-Driven Prognostic Models for GEP-NENs

### 3.1. Registry-Based Studies

Several publications proposing AI models for predicting the prognosis of GEP-NENs have been based on registry data. These studies benefit from large patient cohorts, providing the statistical power needed to develop survival models that generalize across different clinical settings. However, despite their sample size and representativeness strengths, they typically lack highly specific parameters, such as radiomic and molecular features, and often show limited or absent external validation. The registry-based studies about this topic, published in the last five years, are summarized in Table 1 [15,16,17,18,19,20,21,22,23,24,25]. Some studies also demonstrate how algorithm benchmarking can guide optimal model selection depending on the clinical endpoint (e.g., survival, occurrence of metastasis).

Prognostic models predicting the risk of distant metastases have been developed by two research groups [15,16]. Li et al. [15] analyzed data from 1998 PanNETs retrieved from the Surveillance, Epidemiology, and End Results (SEER) registry, alongside an external Chinese cohort. Feature selection and nomogram construction employed Least Absolute Shrinkage and Selection Operator (LASSO), random forest, logistic regression, and Cox regression methods. The resulting nomogram stratified patients according to the risk of liver metastases, achieving an area under the curve (AUC) of 0.88 on internal validation and 0.89 on external validation, along with a concordance index (C-index) of 0.76 for overall survival (OS) prediction.

Following a related methodology, Liu et al. [16] focused on predicting lymph node metastases in gastric NENs using data from 1256 patients (1137 from SEER and 119 from an external Chinese dataset). Their comparative analysis of six different algorithms identified random forest as the most effective, yielding an AUC of 0.81, an accuracy of 0.78, and a specificity of 0.82. Although the results are encouraging, both are subject to several potential sources of bias, including their retrospective design [15], incomplete clinical data such as grading and therapy details [15], limited external validation [15,16], and the absence of integrated radiomic or genomic information [15,16].

In a different approach, Clift et al. [17] applied AI to facilitate the early diagnosis of small bowel NETs in the primary care setting, a particularly challenging context. Their model was trained on data from 11.7 million individuals captured in the Optimum Patient Care Research Database between 2000 and 2023, among whom 382 were diagnosed with small bowel NETs. Model development combined logistic regression, penalized regression, and eXtreme Gradient Boosting (XGBoost). Despite requiring minor calibration (slope 1.16), the XGBoost model outperformed others, achieving an AUC of 0.87, underscoring its potential utility in facilitating earlier clinical recognition of these relatively rare tumors. However, the results should be interpreted with caution, as small bowel NETs represented only 0.003% of the dataset, highlighting the need for larger validation studies before this model can be considered for clinical use.

Other studies have explored the use of AI to predict survival outcomes in patients with GEP-NENs. Liu S. et al. [18] undertook one of the most ambitious AI applications in this field to date, assembling the largest AI dataset in the field on 43,444 gastrointestinal NENs diagnosed between 1992 and 2018 from the SEER registry. A head-to-head comparison of eleven survival prediction algorithms identified the “Oblique Random Survival Forest” as the most accurate, with a C-index of 0.86. Of note, age, histology, and metastatic stage were more influential predictors of survival than treatment-related factors.

Cheng et al. [19] focused their analysis on rectal NETs, utilizing a dataset of over 10,000 SEER cases supplemented by 68 patients from a Chinese hospital. They trained and evaluated six ML classifiers to predict 5-year OS. XGBoost delivered the best performance, maintaining high accuracy across internal and external validation subsets (AUC 0.87) and outperforming the American Joint Committee on Cancer (AJCC) staging system. The most predictive features included age, tumor size, summary stage, and the performance of surgical intervention, highlighting the value of integrating both demographic and therapeutic variables for more accurate risk stratification.

Wu et al. [20] examined a dataset of 714 patients with colorectal NECs from the SEER registry, supplemented by 47 patients from two Chinese centers. Their use of ensemble feature selection methods, including LASSO, random forest, and XGBoost, enabled the identification of six key variables influencing survival. These comprised metastatic status, surgery, radiotherapy, and chemotherapy. Their interactive nomogram achieved time-dependent AUCs of approximately 0.80, highlighting the prognostic relevance of nodal log-odds and multimodal therapy. A web-based calculator was made available to encourage its use in clinical decision-making.

In another noteworthy study, Jiang et al. [21] trained a DeepSurv model to predict 5- and 10-year OS in 3239 PanNEN cases using SEER data. This model incorporated surgical resection, tumor grade, age, chemotherapy administration, and local extension variables. DeepSurv outperformed classical Cox regression, neural multitask logistic models, and random survival forests, achieving AUCs of 0.87–0.90. The model was also deployed as an open-access web tool, allowing clinicians to input individual patient characteristics and obtain real-time survival predictions; however, previous external validation had not been performed in the original study.

Murakami et al. [22] focused on predicting recurrence-free survival (RFS) in 371 patients with non-functional G1/G2 PanNETs who underwent curative resection across 22 Japanese centers (1987–2020). Their random survival forest model outperformed traditional Cox regression, achieving AUCs between 0.73 and 0.83. Notably, non-linear Ki-67 expression and tumor size greater than 20 mm emerged as key determinants of recurrence risk. However, the long enrollment period (over 30 years) may introduce bias due to changes in diagnostic imaging modalities, histopathological classification, and treatment standards.

As far as gastric NENs are concerned, Liu W. et al. [23] applied a Random Survival Forest model to a combined cohort of 286 SEER patients and 92 external cases from China. Their goal was to predict disease-specific survival following surgical resection for gastric NENs. Their model, incorporating conventional clinicopathological factors and the lymph node ratio (LNR), outperformed Cox regression and AJCC staging (external C-index 0.77), confirming the prognostic importance of LNR in these patients (higher LNR linked to worse outcome).

Also, Liao et al. (2024) [24] focused on gastric NENs, applying 10 ML algorithms to predict OS based on 11 clinical variables, including demographic, treatment, and staging factors. Best performance was achieved by random survival forest with an AUC of 0.88–0.96, outperforming the AJCC staging system, but without any external validation of results.

Finally, Ding et al. [25] introduced a dynamic approach by integrating conditional survival (CS) analysis with ML to study 654 gastric NECs from the SEER registry. CS represents the probability of surviving an additional “y” years after surviving “x” years post-diagnosis. Their model, based on random survival forest and LASSO regression, produced a nomogram capable of updating survival probabilities over time. Notably, 5-year CS improved from 48% at diagnosis to 94% after four event-free years, demonstrating the relevance of temporal modeling. Age over 60, high tumor grade, and advanced disease stage negatively impacted prognosis, while surgery and chemotherapy were associated with better outcomes. The resulting nomogram (C-index of 0.78) may serve to tailor follow-up strategies and enhance patient counseling, optimizing patient management.

### 3.2. Institutional Cohort Studies

Recent advances in prognostic assessment by AI have also been explored within institutional cohorts, where detailed clinical, pathological, and imaging data can be more comprehensively curated than in registry-based studies. Key investigations published within the past few years are summarized in Table 2 [26,27,28,29,30,31,32].

In the domain of recurrence prediction, Altaf et al. [26] analyzed a multicenter cohort to predict RFS in 473 patients who underwent curative resection for neuroendocrine liver metastases. Their ensemble model combining random forest, GB, and neural network classifiers incorporated clinical and histopathological data and tumor size. The model achieved a testing AUC of 0.71–0.76 for predicting recurrence within 12 months, substantially outperforming conventional logistic regression approaches. Patients classified as high risk experienced markedly worse 5-year RFS (21% vs. 37%) and OS (62% vs. 90%) compared to their low-risk counterparts. A web-based calculator was created from the model to support postoperative surveillance planning and identify patients who might benefit from early initiation of nonsurgical therapies. However, similar to the limitations noted in Murakami’s study [22], this analysis was based on historical data spanning three decades, during which imaging modalities and systemic therapies have evolved significantly, thus potentially affecting the validity of the results.

Prediction of liver metastasis recurrence following curative resection was also explored by Ma et al. [27]. This retrospective study analyzed clinical data, pathology slides, and radiographic images from 163 PanNET patients who underwent R0 resections. A combined approach integrating computational pathomics (Ki-67 hotspots and Morisita-Horn heterogeneity metrics), ResNet-101-based DL radiomics, and a clinical predictor (nerve infiltration) within a nomogram achieved an AUC of 0.96–0.98. Stratification revealed a median RFS of 28.5 months in the high-risk group vs. 34.7 months in the low-risk group. Despite the model’s high performance, limitations such as the relatively small sample size and the heterogeneity of surgical interventions necessitate caution in generalizing the findings.

Centonze et al. [28] took a broader approach, aggregating clinical data from 422 NECs of various origins (colorectal 37%, lung 26%, gastro-oesophageal 20%, pancreas 10%) to investigate prognostic factors and calculate OS. ML served primarily as an exploratory tool to complement Cox regression analyses, highlighting biological drivers such as Ki-67, index > 55% (hazard ratio–HR: 5.5). The study reinforced the idea that ML can stratify biologically aggressive disease even in the absence of radiomic data, supporting patient stratification for platinum-based chemotherapy vs. alternative regimens and reinforcing ongoing debates within treatment guidelines.

In the area of image-based risk modeling, Song et al. [29] implemented a DL radiomics pipeline to assess the likelihood of 5-year recurrence in patients undergoing resection for PanNENs. The methodology involved U-Net-based segmentation of tumors on arterial-phase CT scans, followed by feature extraction using SE-ResNeXt-50 architecture. Their model achieved an AUC of 0.77–0.83 in external validation, which increased to 0.83 when age and the presence of functional symptoms were incorporated. These results underscore the added value of integrating lightweight clinical variables into radiomics workflows, offering a non-invasive and individualized approach to perioperative risk stratification.

Yang et al. [30] explored DL-based radiomics to predict OS in 162 patients with gastric NENs by extracting image features from both arterial and venous phase contrast-enhanced CT scans. When the radiomic output was combined with traditional clinical variables (Ki-67 index, metastatic stage, tumor size), the resulting hybrid model achieved C-index values ranging from 0.71 to 0.86 across different validation subsets. The model was able to stratify patients into distinct prognostic groups with an HR of 3.12 between high- and low-risk categories. These findings support the idea that DL radiomics can augment traditional biomarkers, particularly in tumors with complex vascular architectures.

A similar direction was pursued by Huang et al. [31], who developed a predictive nomogram for PanNENs by integrating contrast-enhanced ultrasound (CEUS) imaging with clinical features. They fine-tuned a SE-ResNeXt-50 convolutional neural network (CNN) pre-trained on the ImageNet database and repurposed it for CEUS images from 104 PanNEN patients. Thirty-one representative CEUS frames per case were extracted and processed through the DL architecture. The model’s predictive power significantly improved when the DL output was integrated with tumor size and arterial enhancement, yielding an AUC of 0.85–0.97. This strategy capitalizes on the ability of CEUS to highlight microvascular perfusion dynamics, which are increasingly recognized as indicators of tumor aggressiveness. Such non-invasive tools may help inform the surgical approach, especially in borderline or anatomically complex cases.

In the context of treatment response, Hasic-Telalovic et al. [32] investigated the prognostic potential of classical ML techniques using routinely collected clinical data from 70 patients with metastatic NETs treated with first-line SSAs. The study tested ten traditional ML algorithms to predict progression-free survival, ultimately identifying a multinomial naïve Bayes model as the most effective, achieving an accuracy of 80%. Age, metastatic burden, and primary tumor site were the most influential predictors. While the model holds promise as a pragmatic bedside decision support tool, its reliance solely on clinical data without incorporating imaging biomarkers or molecular profiling limits its capacity for broader personalization.

### 3.3. Genetic and Molecular Data Studies

AI-based prognostic models that integrate genomic and transcriptomic information are beginning to gain traction in the management of GEP-NENs. These innovative approaches aim to uncover molecular signatures associated with tumor behavior, enabling risk stratification and disease monitoring with greater precision. In contrast to imaging-based models, which capture phenotypic features, molecular tools offer a window into the biological basis of tumor heterogeneity. However, their widespread adoption in clinical practice remains limited due to high financial costs, restricted availability of sequencing infrastructure, and a general lack of correlation between transcriptomic findings and radiologic or clinical outcomes. The most recent studies in this domain are summarized in Table 3 [33,34,35,36].

Otto et al. [33] conducted a comprehensive transcriptomic analysis on 513 GEP-NEN samples using bulk RNA sequencing, applying single-cell-informed deconvolution techniques to estimate the cellular composition of tumor microenvironments. By performing single-cell-informed deconvolution, they constructed a proliferation-agnostic ML classifier capable of predicting tumor grading, differentiation status (NET versus NEC), and survival outcomes. Their model achieved predictive accuracies between 78% and 81%, suggesting that transcriptional data can serve as a valuable complement to conventional grading systems, particularly when histopathology is ambiguous or tissue samples are limited.

Building on the potential of gene expression profiling, Padwal et al. [34] used a multi-run minimal redundancy maximal relevance feature selection method to distill high-dimensional RNA sequencing data from 214 GEP-NEN specimens. Their random forest models yielded two focused gene panels: a five-gene signature (*ALB, SFRP2, PRRX2, LMO3, NKX2-3*) capable of predicting hepatic metastatic potential and a twelve-gene panel that reliably distinguished tumors of pancreatic origin from those arising in the small intestine. Both classifiers demonstrated external validation accuracies above 90%, underscoring the clinical promise of compact, interpretable biomarker sets.

Greenberg et al. [35] focused on resected PanNETs, where accurate assessment of metastatic potential is critical for postoperative management. Employing an ML framework, their model identified an eight-gene panel (*AURKA, CDCA8, CPB2, MYT1L, NDC80, PAPPA2, SFMBT1, ZPLD1*) that predicted metastatic risk with an AUC of 0.88. This model suggests that transcriptomic features at the time of resection can offer valuable prognostic information and may help refine postoperative surveillance strategies or indicate the need for adjuvant systemic therapy in select cases.

Taking a different approach, Kidd et al. [36] evaluated a liquid biopsy-based tool known as NETest, which analyzes expression of 51 genes via real-time PCR to produce a composite disease activity score. This score ranges from 0 to 100 and reflects both tumor burden and biological activity. Applied across a multicenter cohort of 1072 patients with NENs, the proprietary ML algorithm underlying NETest predicted disease progression with approximately 95% accuracy. By avoiding the need for tissue sampling, NETest may offer a minimally invasive option for longitudinal disease monitoring, particularly useful in patients with inaccessible lesions or contraindications to biopsy.

## 4. Challenges to Clinical Translation of AI in GEP-NENs

The integration of AI into the clinical management of GEP-NENs holds great potential to personalize care, enhance diagnostic accuracy, and optimize therapeutic decisions. However, this integration is far from straightforward. The obstacles go beyond algorithmic performance and include technical, ethical, regulatory, and institutional concerns. For AI to truly augment neuroendocrine oncology practice, these multifaceted issues must be addressed systematically.

### 4.1. Heterogeneity and Potential Biases

GEP-NENs represent less than 2% of all gastrointestinal malignancies. Consequently, even the most experienced centers often lack datasets of sufficient size and diversity to train robust DL models. For instance, even the largest cohort by Liu S. et al. [18] might lack power for subtype-specific analyses, such as gastric NECs. Consequently, NEN’s rarity often forces reliance on surrogate endpoints (e.g., Ki-67 cutoffs) rather than survival outcomes in validation, and this may not fully capture the nuances needed to accurately predict outcomes for these rare subtypes. Furthermore, the limited availability of large, annotated datasets exacerbates the risk of overfitting and restricts model generalizability across the spectrum of primary tumor sites, grades, and imaging modalities [37]. Algorithmic bias is an inevitable consequence: when underrepresented subpopulations (whether based on geography, race, tumor subtype, or healthcare access) are excluded from training data, the resulting models may systematically underperform in real-world clinical settings. This bias risks perpetuating existing health disparities and undermining clinical trust. In oncology-AI literature, fewer than 20% of studies report the racial or ethnic composition of their cohorts, making meaningful subgroup audits impossible [38]. For GEP-NENs, this omission is particularly problematic, given the variability in disease incidence, histological spectrum, and access to treatment across different regions and demographic groups. A model trained predominantly on European data may produce inaccurate predictions for patients of African, Asian, or Latin American descent. Addressing this issue requires embedding fairness and equity considerations throughout the AI development lifecycle, from dataset curation to deployment. Mitigation strategies include subgroup performance reporting, inclusive data collection, and co-development with patient advocates [33,34].

Another pitfall is the phenomenon of automation bias, where clinicians defer to algorithmic output even when it conflicts with clinical intuition. Eye-tracking studies have shown that exposure to incorrect AI suggestions can significantly mislead radiologists, causing them to overlook clear abnormalities. To mitigate this risk, medical education must incorporate AI literacy and critical appraisal skills. Clinicians should be equipped not only to interpret AI outputs but also to understand when and how to override them. Transparent user interfaces and performance dashboards may enhance user confidence and foster appropriate trust in AI systems [38,39].

### 4.2. Technical Limits of Current Models

Despite the rapid growth of AI applications in oncology, most GEP-NEN-specific tools have been developed using retrospective, single-center datasets. Fewer than 5% of models have undergone prospective or multicenter validation, limiting their clinical reliability. Furthermore, the predominance of retrospective studies applying AI to predictive models for NENs introduces further biases, which represent a well-documented concern in medical research, especially when diagnostic criteria or therapeutic interventions change over time (e.g., changes in WHO classification, PRRT adoption). Since AI models are trained on historical data, they may be “out of sync” with current clinical practice, potentially limiting their generalizability. Regulatory agencies are beginning to respond. Two pivotal documents in this context are the U.S. Food and Drug Administration’s (FDA) guidance on Predetermined Change Control Plans (PCCPs) and the European Union’s Artificial Intelligence Act (EU AI Act) [40,41]. The FDA’s 2025 final guidance on PCCPs provides a structured approach for manufacturers of AI-enabled medical devices to manage post-market modifications. Recognizing the adaptive nature of AI systems, the guidance allows for certain pre-specified changes to be implemented without necessitating a new marketing submission, provided these changes are detailed in an approved PCCP. This framework aims to balance the need for innovation with the assurance of device safety and effectiveness throughout the product lifecycle. Concurrently, the EU AI Act, which came into force in August 2024, establishes a comprehensive regulatory framework for AI systems within the European Union. Under this act, AI systems used in medical devices are often classified as “high-risk,” subjecting them to stringent requirements. These include the implementation of robust quality management systems, comprehensive technical documentation, conformity assessments, and post-market monitoring. The act emphasizes transparency, requiring clear information on the AI system’s capabilities and limitations, and mandates human oversight to ensure ethical and safe deployment. Together, these regulatory documents underscore the importance of proactive planning and continuous oversight in the deployment of AI in healthcare. They provide a roadmap for developers and manufacturers to navigate the complex regulatory environment, ensuring that AI-enabled medical devices are both innovative and aligned with safety and ethical standards.

From a technical perspective, regarding AI applications to imaging, differences in CT slice thickness, convolution kernels, or PET reconstruction algorithms can disrupt feature extraction and compromise external reproducibility [11]. Furthermore, radiomics and DL models have achieved encouraging accuracy in differentiating PanNET from adenocarcinoma, yet they still fall short of estimating Ki-67 index or distinguishing NET G3 from NECs with the accuracy required to replace biopsy. False reassurance or misclassification can be harmful, as these subtle differences carry major therapeutic and prognostic implications [42]. Otto et al. [33] instead achieved high accuracy in distinguishing NETs and NECs using a transcriptomic classifier, but the biological interpretability of results remains limited since none of the proposed gene panels were mechanistically linked to imaging phenotypes or therapeutic vulnerabilities. Fragmented data ecosystems observed in AI studies also represent a significant limitation, based on critical disconnects between distinct data types: registries (e.g., SEER) lack radiomic/genomic parameters, while institutional cohorts often omit treatment details. This “siloed variable problem” limits the development of truly multimodal AI tools, since AI models often do not capture the full complexity of a patient’s condition. Only a multimodal approach combining radiomic, genomic, and clinical data would pave the way to precision oncology.

### 4.3. Ethical and Financial Issues

The application of AI in rare diseases like GEP-NENs presents ethical dilemmas, especially regarding secondary data use. Employing archival imaging data to train algorithms may transgress the original scope of patient consent, while cloud-based inference raises cybersecurity concerns [43]. Training robust models across centers is complicated by privacy law and the heightened re-identification risk that comes with small, phenotype-rich cohorts. Rare-disease-specific ethics frameworks now recommend dynamic consent, equitable benefit-sharing, and publicly accessible data-use registers to sustain patient trust. As AI models are increasingly used in clinical practice, ethical oversights will be crucial to ensure that algorithms are transparent, unbiased, and used with appropriate patient consent. Furthermore, while many studies report impressive accuracy metrics, these models often lack prospective validation, meaning they haven’t been tested in real-world clinical workflows, which is a significant hurdle for their clinical adoption.

Financial sustainability also remains a major barrier. The cost-effectiveness of AI tools has not been sufficiently studied, which could hinder their implementation, especially in resource-constrained settings. Implementing AI in clinical practice indeed entails considerable initial and recurring investments ranging from servers, hardware upgrades, staff training, secure cloud storage, software licenses, and cybersecurity audits. Additionally, ongoing maintenance and system updates contribute to the total cost of ownership, posing financial challenges, particularly for institutions with limited resources. Smaller healthcare providers may struggle with the initial investment, and there is a risk that AI could exacerbate existing disparities if not implemented thoughtfully. Ensuring equitable access to AI technologies is crucial to prevent widening the gap between well-funded institutions and those with fewer resources [44]. Despite these initial costs, well-designed AI systems have shown potential to reduce long-term expenditures by improving screening efficiency, decreasing diagnostic delays, and enabling more targeted interventions. A growing body of literature supports the cost-effectiveness of AI in oncology when deployed within thoughtfully integrated workflows and complemented by clinician oversight [45]. Ensuring equitable access to such tools will be essential to avoid deepening health disparities and to realize AI’s promise in improving outcomes for all patients with GEP-NENs.

## 5. Conclusions

AI is steadily reshaping the clinical management of GEP-NENs, offering enhanced diagnostic precision, more nuanced prognostic insights, and the promise of tailored therapeutic approaches. As illustrated by the studies reviewed in this manuscript, AI applications span a broad continuum, from population-scale registries to single-institution cohorts, each contributing unique strengths and limitations to the current evidence base.

Large datasets such as SEER offer statistical robustness and broad applicability, facilitating the development of generalizable survival models. However, these registries often lack integration with imaging or molecular data. In contrast, institutional datasets frequently include high-resolution radiologic and histopathologic features but are typically limited by smaller sample sizes. In addition to these observations, further limitations of the available studies derive from retrospective design and lack of external validation.

Looking ahead, collaborative prospective efforts across institutions are essential. These should prioritize standardized imaging acquisition, integration of molecular and transcriptomic profiles, and inclusion of treatment strategies. Moreover, rigorous cost-effectiveness analyses will be critical to ensure sustainable implementation. Emerging models built on these foundations could support real-time prognostic assessment and therapeutic planning, including causal inference frameworks to inform surgical versus systemic treatment decisions, bridging to molecular-guided tools for refractory cases, and providing a more personalized, data-driven care paradigm for GEP-NEN patients.

## Figures and Tables

**Table 1 cancers-17-01981-t001:** Summary of AI-Driven Prognostic Models for GEP-NENs Derived from Registry-Based Studies.

Study	Aim	Overall Population	AI Technique	Variables Included in the Model	Key Results	Limitations
Cheng 2021 [19]	Prediction of 5-yr OS	SEER-based 10,580 rectal NETs + 68 Chinese cases	6 ML algorithms (SVC, Nu-SVC, Random-Forest, AdaBoost, NB, XGBoost)	Gender, age, race, histologic type, tumor size, tumor number, summary stage, and surgical treatment	- Best performance by XGBoost - AUC: 0.86–0.90	- Retrospective design - Limited external validation - No medical treatment details - No data on radiomics, genomics
Jiang 2023 [21]	Prediction of 5- and 10-yr OS	SEER-based 3239 PanNENs	DeepSurv neural network vs. NMLTR, random survival forest, Cox model	Gender, age, marital status, race, primary site, grade, surgery, chemotherapy, tumor size, and tumor extension	- AUC: 0.87 (5-yr) 0.90 (10-yr) - Web calculator provided	- Retrospective design - Only registry data - No external validation - No treatment details - No data on radiomics, genomics
Li 2023 [15]	Prediction of metastases and OS	SEER-based 1998 PanNETs + 245 Chinese cases	LASSO + Random-Forest feature selection → logistic and Cox nomogram models	Diagnostic model: grade, N-stage, surgery, chemotherapy, tumor size, bone metastasis Prognostic model: subtype, grade, surgery, age, brain metastases	- Nomogram outperforms TNM staging system - AUC: 0.88–0.89 - C-index: 0.76	- Retrospective design - Limited WHO grading - No therapy details - Limited external validation - No data on radiomics, genomics
Liu 2023 [16]	Prediction of lymph node metastases	SEER-based 1137 gastric NENs + 119 Chinese cases	6 ML algorithms: logistic regression, random forest, DT, NB, SVM, and k-NN	Gender, primary site, tumor size, differentiation grade, T stage, M stage	- Best performance by Random Forest - AUC: 0.81 - Accuracy: 0.78	- Mixed retrospective/prospective design - Limited external validation - No data on radiomics, genomics
Murakami 2023 [22]	Prediction of RFS	Multicenter (Japanese registry), 371 PanNETs G1/G2	Random Survival Forest vs. Cox model	Tumor size, Ki-67, residual tumor status, WHO grade, lymph node metastases	- Best performance by Random Survival Forest - AUC: 0.73–0.83 - C-index: 0.84	- Retrospective design - No external validation - Long enrollment period (33 years) - No data on radiomics, genomics
Clift 2024 [17]	Prediction of small bowel NET diagnosis in primary care	UK-registry-based 11.7 million patients, 382 small bowel NETs	Logistic regression, LASSO and ridge logistic models, XGBoost	Age, family history, symptoms or signs (e.g., abdominal pain), data for differential diagnoses (e.g., imaging or coeliac testing)	- Best performance by XGBoost - AUC: 0.87 - Calibration slope: 1.16	- Retrospective design - Extreme rarity (0.003%) in the dataset with modest PPV - No data on radiomics, genomics
Liao 2024 [24]	Prediction of OS	SEER-based 775 gastric NENs	10 ML algorithms (LASSO, random survival forest, elastic net, Ridge, Cox boost, stepwise Cox, SVM, generalized boosted regression modeling, supervised principal component analysis, Cox partial least squares regression)	Gender, age, race, marital status, differentiation, stage, chemotherapy, radiation	- Best performance by random survival forest - AUC: 0.88–0.96	- Retrospective design - No external validation - No data on radiomics, genomics
Liu S. 2024 [18]	Prediction of OS	SEER-based 43,444 gastrointestinal NENs	11 ML algorithms	Gender, age, race, histology, grade, metastases, size, site, tumor number, surgery, N stage, nodes metastases, and removed	- Best performance by Oblique Random Survival Forest - C-index: 0.86 - AUC: 0.87	- Retrospective design - No external validation - No data on radiomics, genomics
Liu W. 2024 [23]	Prediction of DSS after resection (role of LNR)	SEER-based 286 gastric NENs + 92 Chinese cases	Random Survival Forest vs. Cox model	Gender, age, race, marital status, primary site, histology, size, stage, LNR, surgery details, radiotherapy, chemotherapy	- Best performance by Random Survival Forest - C-index: 0.77	- Retrospective design - Limited external validation - No data on radiomics, genomics
Ding 2025 [25]	Prediction of CS	654 SEER NECs	Random Survival Forest + LASSO-Cox nomogram	Age, grade, tumor stage, surgery, chemotherapy	- AUC: 0.87 at 5 years - 5-yr CS rise in 4 years (48– 94 %)	- Retrospective design - Missing data from the registry - No data on radiomics, genomics
Wu 2025 [20]	Prediction of survival	SEER-based 714 colorectal NECs + external 47 Chinese cases	LASSO model, Random Forest, XGBoost feature selection	Gender, age, race, marital status, M stage, log odds of positive nodes, surgery, radiotherapy, chemotherapy, genetic mutations (including *TP53*)	- C-index: 0.65–0.83 - AUC ≈ 0.80 - Web calculator provided	- Retrospective design - Limited external validation - No data on radiomics

Objective: To evaluate the predictive accuracy of AI models using large-scale registry data. Key Metrics: C-index, area under the curve (AUC), accuracy rates, and survival predictions.

**Table 2 cancers-17-01981-t002:** Summary of AI-Driven Prognostic Models for GEP-NENs from Institutional Cohort Studies.

Study	Aim	Overall Population	AI Technique	Variables Included in the Model	Key Results	Limitations
Song 2021 [29]	Prediction of 5-yr RFS	Multicenter, 74 Pan-NENs	U-Net segmentation + DL radiomics (SE-ResNeXt-50) + SVM	Arterial-phase DLR features + age, neuroendocrine symptoms	AUC: 0.77–0.83	- Retrospective design - Small sample size - Limited external validation - No data on genomics
Telalovic 2021 [32]	Prediction of PFS during SSA	Single-center, 74 metastatic GEP-NETs	10 ML algorithms (logistic regression, DT, random forest, SVM, NB, multinomial NB, k-NN, GB, extremely randomized tree classifier, multilayer perceptron)	Gender, age, functioning NET, Ki-67/grading, primary site, metastatic sites, SSA type, adverse events	- Best performance by NB - Accuracy: 80% - Key drivers: age, primary site, number of metastatic sites	- Retrospective design - Small sample size - Limited external validation - No data on radiomics, genomics
Huang 2022 [31]	Preoperative prediction of aggressiveness	Single-center, 104 PanNENs	DL probability from CEUS: Fine-tuned SE-ResNeXt-50 CNN + multivariate logistic nomogram	CEUS images, tumor size, arterial enhancement level	AUC: 0.85–0.97	- Retrospective design - Small sample size - No external validation - Heterogeneous data (CEUS machines) - Surrogate aggressiveness definition (G3 or invasion) - No data on genomics
Centonze 2023 [28]	Identification of prognostic factors	Multicenter, 422 NECs	Random Survival Forest (with Cox comparison)	Gender, age, pure neuroendocrine morphology or mixed, pathological tumor staging, IHC (Ki-67, SSTR2A, p53 and rb1), mutation analysis (*TP53, RB1, KRAS* and *BRAF* genes)	- HR: 5.5 (Ki-67 ≥ 55%) - Additional independent factors: morphology, stage III–IV, primary site (colorectal and gastro-oesophageal worst)	- Retrospective design - No external validation - No treatment details - Heterogeneous data (different primary sites) - No data on radiomics
Yang 2023 [30]	Prediction of OS	Multicenter, 162 gastric NENs	DL radiomics (ResNet-50 DL feature extraction) + Cox nomogram	Arterial and venous DL signatures, Ki-67, tumor longest diameter, metastases	- C-index: 0.71–0.86 - HR: 3.12 (high risk)	- Retrospective design - Small sample size - Manual 2-D segmentation; - CT scanner heterogeneity - No data on genomics
Altaf 2024 [26]	Prediction of early recurrence (≤12 mo) after resection of liver metastases	Multicenter, 473 metastatic NENs	Ensemble AI model	Gender, smoking status, tumor size, number of metastases, bilobar pattern, tumor differentiation, lymphovascular/perineural invasion	- AUC: 0.71–0.76 - Web calculator provided	- Retrospective design - Limited external validation - Long enrollment period (30 years) - No data on radiomics, genomics
Ma 2024 [27]	Prediction of postoperative liver metastasis	Single-center, 163 PanNETs	Integrated nomogram (Pathomics logistic score + ResNet-based DLR + nerve infiltration)	Gender, age, tumor site in the pancreas, vascular/nerve infiltration, stage, *ATRX*/*DAXX*, Ki-67 hotspot index, MH index, DLR score	- AUC: 0.96–0.98 - C-index: 0.96	- Retrospective design - Small sample - No external validation - Heterogeneous surgical scenarios - No data on genomics

Objective: To assess the performance of AI models developed from single- or multi-institutional cohorts. Key Metrics: C-index, area under the curve (AUC), accuracy rates, hazard ratios, and survival predictions.

**Table 3 cancers-17-01981-t003:** Summary of AI-Driven Prognostic Models for NENs Utilizing Genomic and Molecular Data.

Study	Aim	Overall Population	AI Technique	Variables Included in the Model	Key Results	Limitations
Otto 2023 [33]	Prediction of grading, NEC/NET status, and disease-related survival	Multicenter, 513 GEP-NENs	ML for transcriptomic deconvolution (Bioinformatics)	Gene expression data, including *MKI67* expression	- Accuracy: 81% (grading), 78% (NEC/NET) - r: 0.45 (survival)	- Retrospective design - No prospective clinical validation - Complex multi-step workflow - No data on radiomics
Padwal 2023 [34]	Prediction of liver metastases or primary site	214 GEP-NETs	7 ML models (Linear Discriminant Analysis, Random Forest, Classification and Regression Tree, k-NN, SVM, XGBOOST, GBM)	Multiple gene panel (including *ALB, SFRP2, PRRX2, LMO3, NKX2-3*)	Accuracy: 93%–100%	- Retrospective design - No clinical data (including therapy details) - No data on radiomics
Greenberg 2024 [35]	Prediction of metastases after resection	Multicenter, 95 PanNETs	ML applied to define a transcriptomic-based gene panel	Genes: SV2, chromogranin A and B, (*TPH1), ARX, PDX1, UCHL1*, novel 8-gene panel (*AURKA, CDCA8, CPB2, MYT1L, NDC80, PAPPA2, SFMBT1, ZPLD1*)	AUC: 0.88	- Retrospective design - Small sample size - No external validation - No data on clinical status, radiomics
Kidd 2025 [36]	Prediction of tumor progression	Multicenter, 1,336 patients (1,072 NENs)	Ensemble ML converting data from NETest into activity score	Expression levels of 51 neuroendocrine-related genes	AUC: 0.81 (1 yr)	- Proprietary algorithm - Limited public data - No data on radiomics

Objective: To explore the integration of genomic and molecular biomarkers in AI prognostic modeling. Key Metrics: Gene expression profiles, molecular signatures, predictive accuracy, and area under the curve (AUC).

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
