# Peer review of "Artificial Intelligence for Prognosis of Gastro-Entero-Pancreatic Neuroendocrine Neoplasms"

_cancers, 2025, doi:10.3390/cancers17121981_

Round 1

Reviewer 1 Report

Comments and Suggestions for Authors

The article was nicely written but i would suggest to add some comments or a paragraph on the impact of AI on diagnosis of pancreatic NETs, mentioning also the state of the art in the field (in this regard cite the recent papers: PMID: 36657607  and PMID: 35915956). The main question is: could AI replace biopsy for the diagnosis?

The authors should report a comprehensive and accurate search string that should be reproducible

The authors should add in the supplementary material the PRISMA flowchart

The authors should clearly explained why they did not perform a meta-analysis in this paper. Maybe due to the lack of literature?

Author Response

Reviewer 1

The article was nicely written but i would suggest to add some comments or a paragraph on the impact of AI on diagnosis of pancreatic NETs, mentioning also the state of the art in the field (in this regard cite the recent papers: PMID: 36657607  and PMID: 35915956). The main question is: could AI replace biopsy for the diagnosis?

Re. - We thank the reviewer for this valuable comment. While the present manuscript specifically focuses on AI applications for prognostic modeling in NENs rather than diagnostic purposes, we agree that the diagnostic use of AI in EUS warrants acknowledgment. Accordingly, we have added a brief paragraph addressing this topic in the introduction (lines 70 – 80).

The authors should report a comprehensive and accurate search string that should be reproducible

Re. – In the methods section we have reported the string we adopted (lines 102-103). This is a narrative review, not a systematic review, however we decided to adopt a boolean string with the aim of performing a complete PubMed search.

The authors should add in the supplementary material the PRISMA flowchart

Re. – According to your observation, the PRISMA flowchart has been reported only in the supplementary material, and adapted to the indications by: Page MJ, McKenzie JE, Bossuyt PM, Boutron I, Hoffmann TC, Mulrow CD, et al. The PRISMA 2020 statement: an updated guideline for reporting systematic reviews. BMJ. 2021;372:n71. doi:10.1136/bmj.n71

The authors should clearly explained why they did not perform a meta-analysis in this paper. Maybe due to the lack of literature?

Re. – The present review is a “narrative review”, not a systematic review. Thanks for the comment, we have now clearly stated it in the abstract, introduction and simple summary.

Reviewer 2 Report

Comments and Suggestions for Authors

Gastro-entero-pancreatic neuroendocrine neoplasms (GEP-NENs) represent a challenging disease. Their large heterogeneity limits the possibility to provide accurate risk assessments or standardize the most effective therapies for these patients. In recent years, artificial intelligence (AI), and in particular machine learning approaches, have shown real promise in addressing these complexities. By analyzing large volumes of clinical, imaging, and pathological data, AI-based tools can significantly improve the accuracy of survival predictions and guide more tailored treatment strategies. While early results are encouraging, important limitations remain, since available data stem from small, retrospective datasets, sometimes lacking external validation, and concerns around transparency and ethics still represent an open issue. Addressing these gaps will be key to moving from research applications to practical tools that can support everyday clinical decision-making.

In this review, AUTHORS examine the potential applications of AI to develop effective prognostic models in GEP-NENs, and how these models may help clinicians in predicting survival and optimizing patient management.

The ms is interesting.

However it needs some improvements before being considered for acceptance:

These are my comments:

1) The abstract is not well structured. It only contains background and aim towards the end. Rewrite it by accurately summarizing the sections of the study and inserting mini-titles

2) Insert the key questions that the review intends to answer, before the aim which should be better structured

3) “Cancer” is missing from the search keys. I tried to insert it in the logical OR and 107 more articles came out. This is important and should be checked again.

4) Section 3 should be the results section and 4 the discussion section even if it is not very clear. Reorganize the study in this perspective. As for the results, present a brief summary and introduction of the topics addressed. In the discussion, dedicate yourself to comparisons with other studies, the limitations that emerged in the studies and the limitations of the study itself.

5) Attention. There are some typos and omissions (for example, table 3 does not have a legend)

Author Response

Reviewer 2

1) The abstract is not well structured. It only contains background and aim towards the end. Rewrite it by accurately summarizing the sections of the study and inserting mini-titles

Re. - We appreciate the reviewer's insightful comment. While our abstract adheres to the journal's guidelines for narrative reviews, which do not require mini-titles, we have reorganized the text to enhance its structure and readability, in line with your suggestion.

2) Insert the key questions that the review intends to answer, before the aim which should be better structured

Re. – Open issues regarding the topic are reported at lines 80-92, and according to your suggestions the aims of the review have been better explained at lines 93-97.

3) “Cancer” is missing from the search keys. I tried to insert it in the logical OR and 107 more articles came out. This is important and should be checked again.

Re. – We appreciate the reviewer's insightful comment. Initially, we did not include the term “cancer” in our search strategy, as it is infrequently used in reference to neuroendocrine neoplasms. The standard nomenclature for these tumors, as per WHO classifications, includes terms such as “neuroendocrine tumor,” “neuroendocrine carcinoma,” and “neuroendocrine neoplasm.” In response to your suggestion, we revised our search string to include: ("artificial intelligence" OR AI OR "deep learning" OR "machine learning") AND "neuroendocrine cancer." However, this modification did not yield additional studies meeting our predefined inclusion criteria: exclusive focus on AI applications in GEP-NEN prognosis, and publication within the last five years. Consequently, we have retained our original search strategy, as the revised query did not enhance the comprehensiveness or relevance of our literature retrieval.

4) Section 3 should be the results section and 4 the discussion section even if it is not very clear. Reorganize the study in this perspective. As for the results, present a brief summary and introduction of the topics addressed. In the discussion, dedicate yourself to comparisons with other studies, the limitations that emerged in the studies and the limitations of the study itself.

Re. – We agree with the reviewer that, for a systematic review, adopting the structure typically used for original research articles would have been more appropriate. However, as this is a narrative review, we have followed the formatting guidelines provided by Cancers and aligned our structure with that of previously published narrative reviews in the journal.

5) Attention. There are some typos and omissions (for example, table 3 does not have a legend)

Re. – Thanks for the comment, a title has been added to table 3.

Reviewer 3 Report

Comments and Suggestions for Authors

Dear authors 

The manuscript has been revised in an excellent manner. There are no more corrections.

Reviewer 4 Report

Comments and Suggestions for Authors
  1. The current title is lengthy and overly technical. Consider revising it for clarity and conciseness.
  2. In the abstract, succinctly communicate the approach taken (such as a narrative review with systematic search).

  3. Create a new subsection under the Introduction titled “Objectives and Scope” for the explicit indication that this is a technical review concentrating on Artificial Intelligence. It is not intended as a clinical guideline.

  4. Although reference is made to PRISMA flow diagram (Figure 1), it has not been included: ensure it is incorporated in the document.

  5. Think about outlining critical regulatory documents such as FDA guides and EU AI Act.

  6. The objectives of the study along with its key metrics should be included in the table captions but highlighted in a more readable format rather than only inline.

Author Response

Reviewer 3

  1. The current title is lengthy and overly technical. Consider revising it for clarity and conciseness.

Re. - Thanks, we have proposed a new version of the title, hoping you would prefer it.

  1. In the abstract, succinctly communicate the approach taken (such as a narrative review with systematic search).

Re. – According with your suggestion, we have clearly stated that we performed a narrative review (line 28). The term “systematic” has not been used in order to avoid misunderstandings about the methods.

  1. Create a new subsection under the Introduction titled “Objectives and Scope” for the explicit indication that this is a technical review concentrating on Artificial Intelligence. It is not intended as a clinical guideline.

Re. - In response to your suggestions, we have clarified the aims of the review in lines 93-97. Regarding the manuscript structure, we have followed the guidelines provided by Cancers and modeled it on the format of previously published narrative reviews in the journal.

  1. Although reference is made to PRISMA flow diagram (Figure 1), it has not been included: ensure it is incorporated in the document.

Re. - Thanks, we have added a reference for PRISMA diagram, and modified it accordingly (Supplementary Figure 1).

  1. Think about outlining critical regulatory documents such as FDA guides and EU AI Act.

Re. – Thank you for the comment, we have briefly discussed FDA guides and EU AI Act (lines 398 - 418).

  1. The objectives of the study along with its key metrics should be included in the table captions but highlighted in a more readable format rather than only inline.

Re. – Thanks, we have modified table captions accordingly, adding objectives and key metrics.

Round 2

Reviewer 1 Report

Comments and Suggestions for Authors

The manuscript is OK now. Thank you!

Reviewer 2 Report

Comments and Suggestions for Authors

Authors followed my suggestions.

The ms improved.

There are no further comments.

Reviewer 4 Report

Comments and Suggestions for Authors

No further comments need